

# Variational quantum gate optimization at the pulse level

Sean Greenaway[1], Francesco Petiziol[2], Hongzheng Zhao[3] and Florian Mintert[1,4]

**1** Physics Department, Blackett Laboratory, Imperial College London,
Prince Consort Road, SW7 2BW, United Kingdom
**2** Technische Universität Berlin, Institut für Theoretische Physik,
Hardenbergstraße 36, Berlin 10623, Germany
**3** Max Planck Institute for the Physics of Complex Systems,
Nöthnitzer Str. 38, 01187 Dresden, Germany
**4** Helmholtz-Zentrum Dresden-Rossendorf,
Bautzner Landstraße 400, 01328 Dresden, Germany

## Abstract

**We experimentally investigate the viability of a variational quantum gate optimization protocol informed by the underlying physical Hamiltonian of fixed-frequency transmon qubits. Through the successful experimental optimization of two and three qubit quantum gates the utility of the scheme for obtaining gates based on static effective Hamiltonians is demonstrated. The limits of such a strategy are investigated through the optimization of a time-dependent, Floquet-engineered gate, however parameter drift is identified as a key limiting factor preventing the implementation of such a scheme which the variational optimization protocol is unable to overcome.**



# 1   Introduction

Rapid experimental progress in the development of quantum computers has led to the realization of quantum platforms which are approaching the scales necessary for true quantum advantage [1–5]. However, the inherent noise levels of noisy intermediate-scale quantum (NISQ) devices remains a fundamental limiting factor precluding the attainment of results that cannot be obtained classically [6]. As a result, a large body of research has grown around extending the utility of NISQ devices through error mitigation [7–10] or circuit compilation [11–14] algorithms.

These techniques are typically rooted in the gate-based approach to quantum computation [15, 16], in which algorithms are decomposed into a finite set of fundamental basis gates, usually a two-qubit entangling gate (e.g. CNOT) and arbitrary single qubit rotations. This paradigm is ideal for fault-tolerant quantum computation since it is universal. However, the deep circuits necessitated by gate-based algorithms means that implementation of gate-based computations on NISQ devices are severely impeded by gate noise.

One method by which this limitation could potentially be overcome is through the use of *variational quantum gate optimization* (VQGO) [17, 18]. VQGO seeks to obtain the optimal gate parameters that maximize the fidelity of a target gate through a classical optimization routine. While such a routine can be used to increase the fidelity of standard basis gates, it can also be used to optimize non-standard gates such as two-qubit rotation gates and gates that act on more than two qubits, which would otherwise necessitate decomposition into noisy CNOT and single qubit rotation gates. In this way, VQGO can obtain more efficient gate implementations, increasing the range of computations which can be implemented on NISQ devices.

In order to implement a VQGO routine, a parameterized quantum gate is required. A natural choice for this is to use the native operations for a given device as the control parameters, choosing target gates that can be realized using those operations. In this work, fixed-frequency, fixed-interaction (FF) transmon qubits [19] as implemented by IBM Quantum [20] are used as the basis for the VQGO routine. FF transmons are highly controllable, with individually controllable $ZX$ and $ZY$ interactions and arbitrary single qubit rotations natively available. However, the interaction terms are accompanied by significant noise terms which, alongside experimental imperfections, can severely reduce the fidelity of implemented gates. Thus, the platform is ideally suited to optimization via VQGO. We assess the extent to which VQGO can overcome these limitations and thereby realize high fidelity gates on currently available hardware.

The native entangling operation in FF transmon qubits is the *cross-resonance* gate [21–23], obtained by driving one qubit at the resonant frequency of another to which it is coupled. This results in an entangling $\exp(i\theta ZX)$ operation, with the Rabi angle $\theta$ controlled by the pulse amplitude and duration. In addition to the desired coupling term, unwanted spurious single qubit terms are also generated which must be controlled for high fidelity gates to be realized.

In principle, quantum optimal control schemes [24] based on theoretical models of this interaction can be designed to eliminate these unwanted terms. However, while sophisticated models of the interaction have been developed [25, 26], they cannot be used to make a priori predictions about an experimental system given the susceptibility of experimental system parameters to drift and the limited access to these parameters afforded to end users. As a black-box optimization routine, VQGO can be used to obtain high-fidelity gates without rigorously characterizing the underlying system, making it well-suited to this application.

The choice of target gates for the VQGO routine explored in this work is motivated by the form of the cross-resonance interaction, which we briefly review in Sec. 2. The optimization of a two-qubit $ZX$ gate is presented in Sec. 3, corresponding to the cross-resonance interaction with the error terms eliminated, before an extension to a three qubit gate consisting of two simultaneous cross-resonance interactions is made in Sec. 4. In both cases the VQGO routine is very effective, resulting in the experimental realization of high fidelity gates.

Having demonstrated the utility of VQGO for obtaining high fidelity gates based on time-independent Hamiltonians, we try to generalize the approach to the more challenging application of implementing time-dependent, Floquet-engineered systems [27, 28]. A scheme for realizing a three-body $ZYZ$ gate [29–31] at stroboscopic times is used as the testbed for such an application, with the VQGO results presented in Sec. 5. The VQGO protocol was able to improve the fidelity of the realized gate when compared with unoptimized gate parameters. However, significant parameter drift over the time frame of the optimization poses a severe limitation to this method, preventing the protocol from reaching similarly high fidelities to the other gates.

Our results show that VQGO is effective at obtaining optimal drive routines for novel quantum gates based on static effective Hamiltonians outside the usual set of basis gates. We identify parameter drift as the primary limiting factor preventing VQGO from obtaining similarly high fidelity gates based on time-dependent Hamiltonians. In principle, this means that control schemes that are designed to be robust to parameter drift could be amenable to optimization through VQGO. The use of VQGO could allow for more efficient compilation of quantum circuits than is possible using current gate decomposition approaches, thereby increasing the utility of NISQ devices.

## 2 Experimental system

While the general techniques discussed in this work are applicable to any quantum system, the specific choices of optimization targets and figures of merit are motivated by the experimental system used to perform the optimizations. Here the experimental platform consists of fixed-frequency, fixed-interaction (FF) transmon qubits capacitively coupled together. This is the experimental platform used by IBM in their IBM Quantum systems [20].

These systems may be modelled as a series of $n$ anharmonic Duffing oscillators [32]

$$H^{\text{Duff}} = \sum_{i=1}^{n} (\omega_i a_i^\dagger a_i + \alpha_i a_i^\dagger a_i^\dagger a_i a_i + \sum_{\langle i,j \rangle} J_{ij}(a_i - a_i^\dagger)(a_j - a_j^\dagger)), \tag{1}$$

with anharmonicities $\alpha_i$ and resonant frequencies $\omega_i$. The capacitive coupling strength $J_{ij}$ between nearest-neighbour transmons (represented by the angled brackets) is fixed by the hardware and cannot be externally controlled. As a result, $J_{ij}$ must be sufficiently weak such that, in the absence of driving fields, no entanglement between coupled qubits is generated.

In such a system, dynamics may be induced by driving the system with microwave pulses. If these pulses have amplitudes which are significantly lower than the transmon anharmonicities,

then only the lowest two energy levels will be populated and the dynamics can be accurately described by a simplified qubit model. Restricting Eq. (1) to a qubit model yields

$$H(t) = \sum_{i=1}^{n} \frac{\omega_i}{2} Z_i + \sum_{\langle i,j \rangle} J_{ij} Y_i Y_j + \sum_i D_i(t) X_i \,, \tag{2}$$

where the final sum is over the subset of qubits upon which the pulses are applied and where the driving $D_i(t)$ may be conveniently parameterized in terms of dimensionless pulse envelope parameters $d_i^X(t), d_i^Y(t)$ and $d_i^Z(t)$ as

$$D_i(t) = \mathrm{Re}\left[ \frac{\Omega}{2}\left( (d_i^X(t) - i d_i^Y(t)) \exp\left( -2i \int_0^t d_i^Z(t') dt' \right) \right) e^{i(\omega_i + \Delta_i)t} \right], \tag{3}$$

with $\Delta_i$ the detuning from the resonant qubit frequency. On resonance driving generates single qubit dynamics. In the frame rotating with the qubit frequencies defined by a unitary transformation using the rotation operator $\exp(i\sum_i \frac{\omega_i}{2} Z_i)$, the effective Hamiltonian for such a driving (applied to the $i$th qubit) is

$$\tilde{H}_i(t) = \frac{\Omega}{2}(d_i^X(t) X_i + d_i^Y(t) Y_i + d_i^Z(t) Z_i) \,. \tag{4}$$

In this way, full single qubit quantum control of individual qubits is possible. Entangling operations between pairs of coupled qubits are also able to be generated using off-resonant drive pulses, making transmon qubits highly expressible as a system for quantum computation and simulation. This interaction is outlined in the following section.

## 2.1 The cross-resonance gate

An entangling interaction between two coupled qubits can be generated by driving one qubit at the resonant frequency of the other, resulting in a *cross-resonance* interaction [21–23]. In the frame rotating with the qubit frequencies, the effective Hamiltonian resulting from such a drive is given by

$$H_{ij}^{\mathrm{CR}} = \sum_{A \in \{\mathbb{1}, X, Y, Z\}} c_{\mathbb{1}A} \mathbb{1}_i A_j + c_{ZA} Z_i A_j \,, \tag{5}$$

where the $i$th qubit (the control) is driven at the resonant frequency of the $j$th (the target). The terms in Eq. (5) have different magnitudes due to the fact that the parameters in the drive Hamiltonian Eq. (2) are of significantly different magnitudes: in particular, $J_{ij} \ll \Omega \ll \Delta_{ij}$ with $\Delta_{ij} = \omega_i - \omega_j$. The largest term, the single qubit $Z\mathbb{1}$ rotation on the drive qubit, is proportional to $\Omega_i^2/\Delta_{ij}$ and arises due to a strong AC-Stark shift from the off-resonant drive. Next largest in magnitude are the two qubit $ZX$ and $ZY$ entangling operations and the single qubit $\mathbb{1}X$ and $\mathbb{1}Y$ rotations on the target qubit, all of which are proportional to $J_{ij}\Omega_i/\Delta_{ij}$. These terms arise from the interplay between the non-commuting drive and static coupling terms in Eq. (2). Finally, the single qubit $\mathbb{1}Z$ rotation on the target qubit and the $ZZ$ interaction originate from the weak static coupling and are proportional to $J_{ij}^2/\Delta_{ij}$.

For most purposes, an ideal starting point for experiments using the cross-resonance interaction is a pure $ZX$ interaction. In such a scheme, the other terms can be considered error terms unless stated otherwise. The weakness of the $ZZ$ and $\mathbb{1}Z$ terms arising from the qubit-qubit coupling means that these terms can be neglected, however the rest of the terms must be eliminated experimentally. These remaining error terms can be straightforwardly corrected by adjusting the cross-resonant drive envelope phase and applying additional single qubit control terms, both of which are able to be accurately controlled experimentally. All that therefore remains is to determine the magnitude of these corrections.

As mentioned above, our approach treats the FF transmon system as a black-box and attempts to find optimal parameters through VQGO rather than rigorously characterizing the experimental system to fit the theoretical models [25, 26]. This approach assumes that the experimental errors are dominated by the unitary errors arising from miscalibrated drive parameters. Since only unitary control terms are available, incoherent errors such as decoherence and dephasing cannot be directly corrected by the VQGO routine and thus will necessarily reduce the fidelity of applied gates. The two most significant sources of incoherent errors are decoherence and measurement error. A key advantage of FF transmon qubits is their long coherence times which, at approximately 100 $\mu s$ are much longer than the gate times investigated here, with the longest gate times being less than 5 $\mu s$. Decoherence over these short times is therefore minimal and can be ignored. Measurement error is known to be a significant problem for FF transmon qubits [33]. Since all the figures of merit for the optimizations performed here average over a number of different expectation value measurements, the effect of measurement error is well approximated by unbiased stochastic error. In this case, the optimal parameters for a given gate should remain unchanged by the presence of measurement error, and thus VQGO should still be effective. As a consequence of neglecting measurement error, the obtained fidelities will be lower than expected based on previously published fidelity measurements [34]. For example, the identity gate under this assumption has an experimental fidelity of approximately 95%.

## 3 Optimizing the cross-resonance gate

Having access to a high fidelity entangling operation is highly important for quantum computing platforms. As such, the natural starting point for a VQGO protocol implemented on a FF transmon device is the optimization of a pure $\exp(i\theta ZX)$ gate. Such a protocol is both inherently useful, since applying a pure $ZX$ gate for a rotation angle of $\pi/4$ yields a maximally entangling gate which is equivalent to a CNOT up to single qubit rotations, and highly useful as the starting point for analogue and hybrid quantum computations [35]. Applying an analogue $ZX$ pulse for varying durations can result in improved fidelity in quantum simulations when compared with using CNOT decompositions [36].

Control schemes implemented on FF transmon qubits typically involve implementing an echoed cross-resonance pulse sequence in order to refocus most of the error terms [22, 37]. However, it is possible to directly control the Hamiltonian terms instead. This cuts down on the number of pulses which need to be applied and additionally allows for the simultaneous application of additional control pulses, potentially expanding the utility of the cross-resonance interaction into the fields of quantum optimal control and analogue quantum simulation [38, 39]. The strategy of controlling individual Hamiltonian terms, rather than using an echoed pulse sequence, is employed in this work.

Aside from the choice of target gate, two additional choices must be made for the implementation of VQGO: a figure of merit quantifying the quality of the experimental gate and a classical optimization routine. Different figures of merit are best suited to different gates, and so the choices of figure of merit will be discussed with respect to the different target gates in the subsequent sections. For the classical optimization routine, *Bayesian optimization* (BO) [40], a probabilistic machine learning method is utilized. BO is well suited to applications in which evaluation of the figure of merit incurs a significant overhead due to it requiring, for example, an experiment to be performed and has been previously implemented successfully in various quantum optimal control applications [18, 41–46]. A thorough overview of BO can be found in Refs. [47–49].

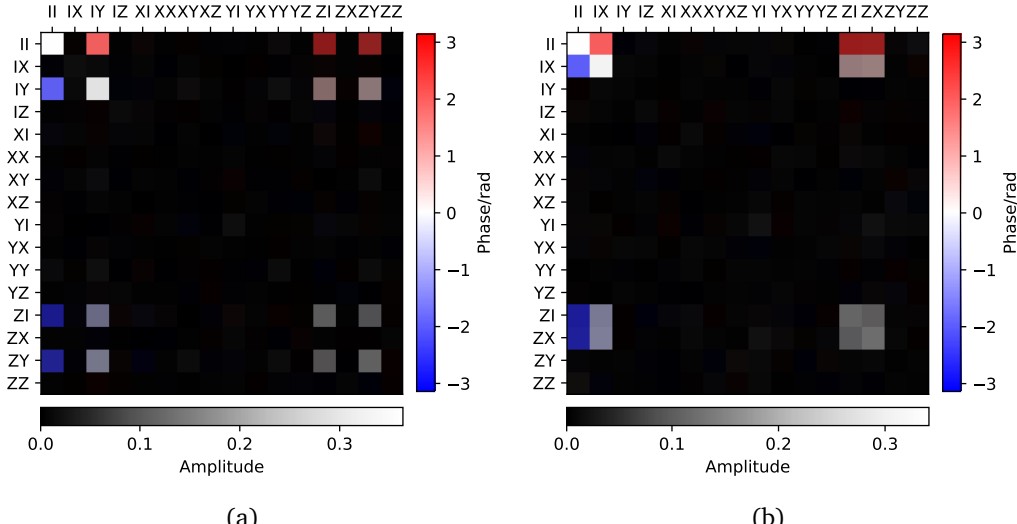

$$\text{(a)} \qquad\qquad\qquad\qquad \text{(b)}$$

Figure 1: Quantum process matrices extracted from the application of a cross-resonance interaction implemented on the `ibmq_paris` quantum device. (a) shows results from applying a drive in which the phase of the drive envelope in Eq. (3) requested by the user is 0, and shows significant phase error due to drive line nonlinearities. (b) shows the same interaction with an additional phase added to the drive envelope optimized to eliminate this error.

## 3.1 Phase calibration

Before working with the cross-resonance gate, it is convenient to first calibrate the phase of the applied cross-resonant pulses such that the experimental effective Hamiltonian matches the expected theoretical terms. For a pure $ZX$ gate, the drive envelope in Eq. (3) should be purely real (i.e. $h^Y = 0$). However, systematic experimental errors such as delays in the drive lines can induce phase shifts on the signal generated by the arbitrary waveguide generator, resulting in a non-zero value of $h^Y$. The drive envelope phase requested by the user therefore must be adjusted to eliminate the $h^Y$ term and ensure the effective Hamiltonian consists only of $ZX$, $Z\mathbb{1}$ and $\mathbb{1}X$ terms.

This phase can be optimized by applying the cross-resonance drive to the $|++\rangle$ initial state and measuring in the $X$ eigenbasis on both qubits. The optimal phase is then found by minimizing the sum of projections into the $|+-\rangle$ and $|--\rangle$ states, which can be straightforwardly achieved using the individual projective measurements for each qubit.

In order to verify that the phase optimization routine successfully eliminates the unwanted terms, an unbiased validation method is required. A natural choice for this is provided by *quantum process tomography*. Quantum process tomography characterizes a quantum process $\Lambda$, which may be written in terms of its action on an arbitrary input state $\rho$ as

$$\Lambda(\rho) = \sum_{i,j=1}^{d^2} \chi_{ij} \sigma_i \rho \sigma_j, \tag{6}$$

where $\{\sigma_i\}$ is the operator basis formed form the $n-$fold tensor products of Pauli matrices. Quantum process tomography is used to experimentally extract the *process matrix* $\chi_{ij}$ which fully characterizes $\Lambda$ [50].

Fig. 1 shows the experimental process matrices for a two qubit cross resonance interaction before and after phase optimization. Prior to phase optimization (Fig. 1a), the dynamics are dominated by terms generated by unwanted $ZY$ and $\mathbb{1}Y$ terms due to the phase misalignment,

however by adjusting the phase of the pulse, these can be virtually eliminated, with the final process matrix almost entirely consisting of terms generated by $ZX$, $Z\mathbb{1}$ and $\mathbb{1}X$ (Fig. 1b).

The remainder of this work will exclusively use drive pulses which the drive envelope in Eq. (3) is either purely real ($h^Y = 0$) or purely imaginary ($h^X = 0$) once this phase misalignment is accounted for. In principle, however, once the calibration phase is known, any two-body interaction consisting of a coherent mixture of $ZX$ and $ZY$ terms can be generated, allowing for a wide array of interaction terms to be implemented, which is another advantage of applying VQGO to FF transmon qubits.

## 3.2 Reduced process tomography

With the phase error from the applied drive eliminated, the effective Hamiltonian arising from the unoptimized cross-resonance interaction consists only of terms $ZX$, $Z\mathbb{1}$ and $\mathbb{1}X$ terms. Since the dynamics generated by such a limited set of Hamiltonian terms is only a small subset of the full Hilbert space, it is possible to describe the action of the gate to a high degree of accuracy using a significantly reduced set of measurements, in a protocol known as *reduced process tomography* [51–53].

Quantum process tomography provides an effective verification that an optimization has succeeded, since it is unbiased and captures the full action of a particular gate. However, performing full process tomography is extremely experimentally expensive, making it impractical for the iterative evaluation of gate fidelity required for VQGO. For certain gates, however, most of the elements of $\chi_{ij}$ are known to be vanishing a priori, meaning that only a reduced subset of measurements is required to evaluate it. This is the underlying idea behind reduced process tomography.

Explicitly, for a general two qubit cross resonance interaction of the form

$$U_{ZX}(t) = \exp(-i(J_{ZX}ZX + J_{Z\mathbb{1}}Z\mathbb{1} + J_{\mathbb{1}X}\mathbb{1}X)t), \tag{7}$$

the dynamics can be entirely captured by performing only a single two-qubit state tomography experiment. The quantum gate defined through the application of Eq. (7) consists only of a weighted sum of operators $\{\mathbb{1}\mathbb{1}, Z\mathbb{1}, \mathbb{1}X, ZX\}$. Each of these operators maps the initial state $|{+}0\rangle$ to one of a set of mutually orthogonal final states as

$$\mathbb{1}\mathbb{1}|{+}0\rangle = |{+}0\rangle\,, \tag{8}$$

$$Z\mathbb{1}|{+}0\rangle = |{-}0\rangle\,, \tag{9}$$

$$\mathbb{1}X|{+}0\rangle = |{+}1\rangle\,, \tag{10}$$

$$ZX|{+}0\rangle = |{-}1\rangle\,. \tag{11}$$

Thus an approximation to the full process matrix can be obtained by performing quantum state tomography on the resulting output state and obtaining the process matrix as the matrix elements of the output density matrix. Having extracted the reduced process matrix $\chi^{\text{red}}$, an approximation to the quantum process fidelity can be made through the overlap between $\chi^{\text{red}}$ and the ideal process matrix $\chi^{ZX}$

$$F \approx \text{Tr}\left[\chi^{\text{red}}(\chi^{ZX})^\dagger\right]\,. \tag{12}$$

This approximation will be referred to as the *reduced $\chi$ overlap*. The quality of the reduced $\chi$ overlap as an approximation to the process fidelity depends on how closely Eq. (7) captures the experimental dynamics. In order to verify this, the reduced $\chi$ overlap and full quantum process fidelity can be evaluated using the same experimental data, extracting the reduced process matrix as a subset of the full tomographic expectation values. Fig. 2 shows the experimental results of such a procedure for 100 gates generated by varying the cross-resonance

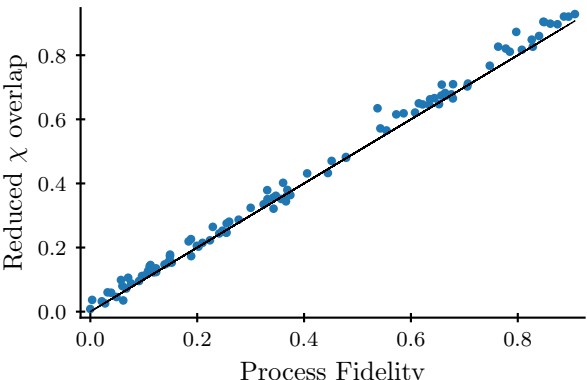

Figure 2: Plot of reduced $\chi$ overlap against process fidelity for 100 experimental gates implemented with varying cross-resonance and single qubit correction pulse amplitudes. Both figures of merit were extracted from the same set of experimental data, with the process fidelity calculated using the full set of 240 expectation values required for full quantum process tomography and the reduced $\chi$ overlap using a subset of the data. The reduced $\chi$ overlap approximates the full process fidelity very well, and should therefore be an efficient alternative to it, requiring only 12 expectation value measurements rather than 240.

amplitude and the amplitudes of compensating $Z\mathbb{1}$ and $\mathbb{1}X$ pulses. Since the range of fidelities are obtained by varying the same parameters as are used in an optimization protocol, the extent to which the reduced $\chi$ overlap approximates the process fidelity is a strong indication of its viability as a figure of merit for VQGO.

In the ideal case, the data should lie entirely along the diagonal of Fig. 2. All of the data are indeed very close to this diagonal, which is shown as a black line. Moreover, the deviations are consistent with the level of measurement error in the IBM Quantum devices. Given the significant reduction in overhead from using the reduced $\chi$ overlap over evaluations of the full process fidelity, a reduction from 240 expectation value measurements per evaluation to just 12, the quality of the reduced $\chi$ overlap as an approximation makes it an appropriate choice as the figure of merit for an optimization.

## 3.3 Obtaining an optimal cross-resonance interaction

Having obtained a figure of merit which can efficiently evaluate the quality of an experimental cross-resonance gate, a VQGO routine can be implemented using BO as the classical optimizer and using the amplitudes of the cross-resonance single qubit correction pulses as control parameters. The target gate for this optimization is a pure $ZX$ interaction at the maximally entangling Rabi angle of $\theta = \pi/4$. The control pulses over which the optimization was performed were based on Eq. (3) and were parametrized as follows:

$$D_{12}(d_1^{ZX}, d_2^X, d_1^Z; t) = \mathrm{Re}\left[\frac{\Omega}{2}\left(d_1^{ZX}e^{i\phi_{ZX}}e^{i\omega_2 t} + d_2^X e^{i\omega_2 t}\right) + \underbrace{\exp\left(-2i\int_0^t d_i^Z(t')dt'\right)}_{\text{Virtual } Z \text{ rotation}}\right], \quad (13)$$

with the control parameters being the amplitudes of the cross-resonance and single qubit $\mathbb{1}X$ pulses ($d_1^{ZX}$ and $d_2^X$ respectively) and the magnitude of the single qubit $Z$ rotation on the target qubit, which was indirectly implemented as a virtual $Z$ gate [54] by updating the qubit's resonant frequency in software. The pulses were also multiplied by a Gaussian ramp up/ramp

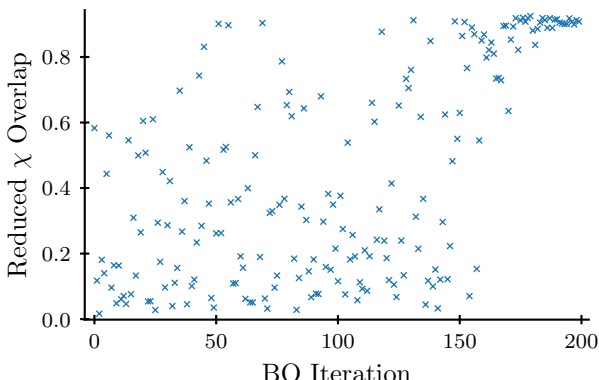

Figure 3: Convergence plot for the optimization of the cross-resonance gate, showing the overlap between the ideal process matrix and the reduced process matrix as a function of the Bayesian optimization iteration. The optimization explores a range of parameter values, finding some high fidelity points before converging to a peak value of 0.93, demonstrating the success of the optimization.

down to avoid discontinuous pulses, however the parameters for these ramps were kept consistent throughout the optimization. Fig. 4 shows a plot of the amplitudes $d_1^{ZX}$ and $d_2^X$ as a function of the pulse envelope for the optimal set of pulse parameters obtained using VQGO, with the top (yellow) plot showing the $d_1^{ZX}$ amplitude multiplied by the phase to compensate for phase error (with the lighter shade the real part and the darker shade the imaginary part and the bottom (blue) plot showing the $d_2^X$ amplitude.

Fig. 3 shows the reduced process matrix overlap as a function of the BO iterations for such an optimization. In the first 150 iterations of the optimization, the optimizer explores the parameter space, thus there are many evaluations which are of poor fidelity. In the latter stage of the optimization, however, the optimizer has enough information to converge to a point at which the reduced process matrix overlap is maximized, with all evaluated points being above overlaps of 90%.

While the quality of the channels realized by the pulse scheme can vary significantly as shown by Fig. 3, the actual pulses used to implement them look very similar, differing only by the magnitude of the applied pulse amplitudes. Due to the non-trivial transformations of the signal that occur in the physical experiments, meaning that the qubits do not experience the ideal pulse as requested by the user, it would be extremely difficult to tell *a priori* what the optimal pulse parameters should be to realize a high fidelity gate. By using VQGO, this difficulty can be side-stepped, allowing for high fidelity pulse schemes to be obtained without necessitating a rigorous characterization of this transformation.

The quality of the final parameters found by the optimization routine is shown through the full process matrix evaluated using process tomography in Fig. 5(a), with Fig. 5(b) showing the same process matrix with the largest elements set to 0 and with the color bar rescaled to show the magnitude of the small residual error terms. The target Rabi angle for this optimization was $\pi/4$, corresponding to a maximally entangling gate with four equal-magnitude process matrix elements, $\chi_{\mathbb{1}\mathbb{1},\mathbb{1}\mathbb{1}} = \chi_{ZX,ZX}$ and $\chi_{\mathbb{1}\mathbb{1},ZX} = -\chi_{ZX,\mathbb{1}\mathbb{1}}$, which is realized to very high fidelity (93%) in the final process matrix. As mentioned above, while this is significantly lower than the reported CNOT error rates [20], much of this reduction can be attributed to measurement error. In order to fairly compare this result to the state-of-the art method in the presence of this measurement error, the process matrix for the IBM-calibrated $ZX$ gate was experimentally extracted. This was obtained by taking the CNOT gate pulse sequence and stripping out the single qubit rotation gates used to convert the native $ZX$ gate to a CNOT. The resulting pulse

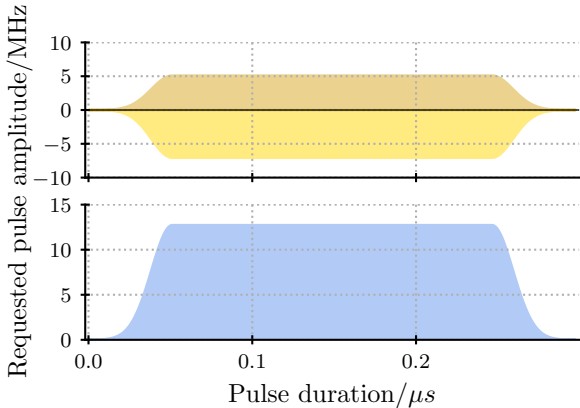

Figure 4: Plot showing the requested drive pulse amplitudes used in the optimization of the single application of the cross-resonance drive outlined in Sec. 3.3, with the yellow plots corresponding to cross-resonance pulses and the blue plots corresponding to the single qubit resonant pulses needed to counteract the spurious single qubit $\mathbb{1}X$ term from the cross-resonance interaction and where the light and dark portions of the plots correspond to the real and imaginary components of the pulses respectively. This set of pulses corresponds to the highest fidelity observed in the optimization. The phase of the cross-resonance drives is non-zero due to the phase calibration outlined in Sec. 3.1. The $Z\mathbb{1}$ term may be corrected using a virtual $Z$ rotation [54] and so is not represented in this plot of physical pulses.

sequence yields a process fidelity of 93%, matching the one obtained in this work, but achieves so at the price of using multiple pulses and longer total pulsing time.

The fact that the final $ZX$ gate optimized through VQGO matches the fidelity achieved by IBM indicates that the optimization routine performed as well as it could have done – i.e. the obtained fidelity is as high as can be achieved using the control scheme presented in this work. Additionally, the fidelity of the IBM-calibrated CNOT gate was also evaluated, which yielded a fidelity of 92.8%. This implies that the optimized $ZX$ gate could also be used to generate a CNOT gate with a comparable fidelity. However, such an application is not the most useful application of the optimization procedure. The key advantage of using VQGO over the IBM definition lies in its flexibility – it can be used to implement gates which cannot be natively implemented using the standard IBM pulse definitions and 'hardware' interactions. This is illustrated in the following section by application of the protocol to a more complicated three-qubit gate.

## 4  Optimizing non-commuting interactions

Having demonstrated the utility of pulse-level VQGO on the cross-resonance gate, a natural extension is made to a three qubit quantum gate. For the $ZX$ optimization, all of the terms which generate Eq. (7) mutually commute, meaning the control landscape can be factorized into a product of individual control terms for the $ZX$ and single qubit $Z\mathbb{1}$ and $\mathbb{1}X$ pulse amplitudes, greatly simplifying the optimization. Additionally, the favorable structure of the process matrix that permits the definition of the reduced $\chi$ figure of merit is not generic for all gates that can be implemented based on the cross-resonance gate. As such, it is pertinent to evaluate the viability of pulse-level VQGO when the target gate lacks the convenient features of the $ZX$ gate.

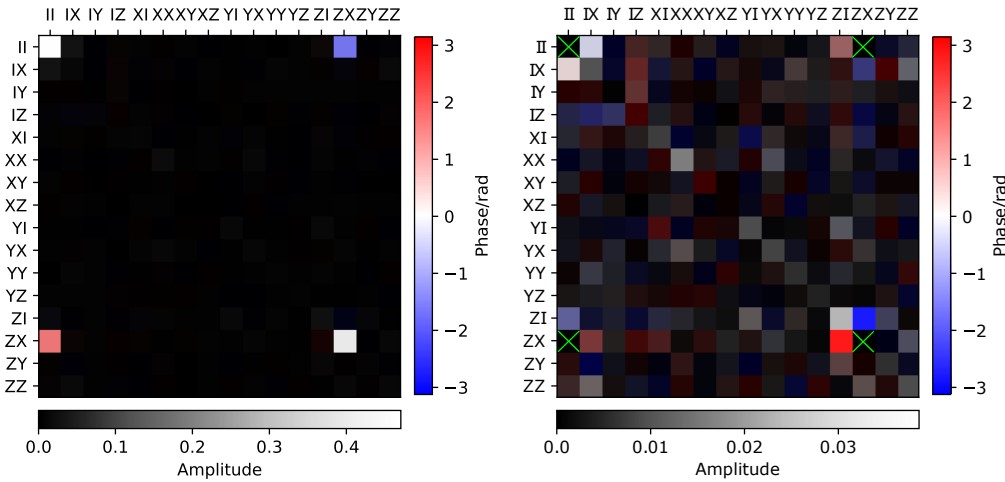

(a) Experimental process matrix

(b) Experimental process matrix (a) with largest elements dropped and rescaled to show small elements

Figure 5: (a) Quantum process matrix extracted from the application of a cross-resonance interaction optimized for the gate $\exp(i\pi/4ZX)$ using Bayesian optimization and implemented on the `ibmq_paris` quantum device. The ideal process matrix consists of the four terms which dominate the optimized process matrix, each with magnitude 1/2. (b) The same optimized process matrix as (a), but with the four largest elements (corresponding to $\chi$ elements indexed by products of $\mathbb{1}\mathbb{1}$ and $ZX$) set to 0 and the color bar rescaled to show the magnitude of the less significant terms. The optimized process matrix has some small residual unwanted terms and the magnitudes are slightly suppressed, but nevertheless yields a fidelity of approximately 0.93 even without any measurement error mitigation.

A natural choice for a three qubit target gate in such a setting is the unitary evolution operator generated by a Hamiltonian of the form

$$H = J\left(ZX\mathbb{1} + \mathbb{1}YZ\right), \tag{14}$$

which may be implemented using only constant pulses based on the native cross-resonance operations, with a purely real ($h^Y = 0$ in Eq. (3)) cross-resonance pulse on the first qubit and a purely imaginary ($h^X = 0$ in Eq. (3)) pulse on the third qubit. Although the implementation of gates based on Eq. (14) appears similar to the $ZX$ gate optimized in the previous section, the error terms generated by the two unoptimized cross-resonance drives do not mutually commute, significantly complicating the optimization landscape. Additionally, since this is a three qubit gate, the number of parameters over which an optimization may be performed is doubled, providing an additional complication.

Rather than optimizing the full gate starting from a completely unoptimized set of pulse parameters, the constituent $ZX\mathbb{1}$ and $\mathbb{1}YZ$ interactions can first be optimized to find the optimal cross-resonance drive amplitudes and correction rotations for counteracting the AC Stark shift effects. In principle, this method should also yield the optimal single qubit $\mathbb{1}X\mathbb{1}$ and $\mathbb{1}Y\mathbb{1}$ pulse amplitudes, however the values obtained for the two-body interaction are not necessarily optimal for the three qubit $ZX\mathbb{1} + \mathbb{1}YZ$ gate. The control parameters used in the optimization of this gate were the amplitudes of the two cross-resonance pulses and of the single qubit resonance pulse, the magnitudes of the virtual $Z$ rotations [54] on the drive qubits for the cross-resonance pulses and the phase of the single qubit resonance pulse.

For each two-body interaction, all of the error terms mutually commute. Thus, the different terms that generate Eq. (7) can be factorized. Similarly, for the $\mathbb{1}YZ$ term an analogous factorization of the terms which generate the unitary

$$U_{YZ}(t) = \exp\left(-i\left(J_{YZ}YZ + J_{\mathbb{1}Z}\mathbb{1}Z + J_{Y\mathbb{1}}Y\mathbb{1}\right)t\right),\tag{15}$$

may be made. For each two-body interaction, there are therefore an infinite number of solutions for each of the parameters of the form $\theta_{\text{opt}} + m2\pi$, where $\theta_{\text{opt}}$ is the parameter which exactly realizes the desired operation with no over or underrotation. As the cross-resonance interaction is weak, a significant change in the drive amplitude is required to enact a moderate change in the effective $ZX$ strength. It is therefore possible to constrain the optimization domain such that the applied drive amplitudes only span a single Rabi oscillation.

For the single qubit correction terms, this is not straightforward to achieve. This is due to the fact that the smallest possible amplitude may still yield a non-zero effective $\mathbb{1}X$ term. Additionally, since the resonant fields are much stronger than the cross-resonance interaction, the required compensating drives need to be applied at very low amplitude. At such low amplitudes, nonlinearities in the resonance drive lines can result in unwanted phase errors, meaning that the exact cancellation amplitude may not be optimal.

A solution to this is to optimize the correction pulse on the central qubit separately once the two-body interactions have been optimized – that is, the magnitudes of the pulses which realize the $ZX\mathbb{1}$ and $\mathbb{1}YZ$ interactions, as well as the compensating $Z\mathbb{1}\mathbb{1}$ and $\mathbb{1}\mathbb{1}Z$ correction rotations are individually optimized before the optimal single qubit $\mathbb{1}X\mathbb{1} + \mathbb{1}Y\mathbb{1}$ pulse amplitude is optimized. Only the amplitude which exactly cancels the single qubit terms will yield maximum fidelity, thus there will only be one optimal solution. By optimizing both the amplitude and the phase of the applied drive pulse, the phase errors can be simultaneously corrected.

While this requires an increase in overhead, the protocol can be scaled to large system size by optimizing blocks comprising a small number of qubits separately and making use of parallelization to simultaneously optimize interactions which are physically distant enough that cross-talk is unlikely. This would necessitate an increase in quantum resources by only a constant factor dependent on the target problem and the geometry of the experimental system and a linear increase in classical computational resources. The latter could also, in principle, be reduced through information-sharing protocols [44].

## 4.1 Zero-fidelity estimation

Unlike the two-qubit $ZX$ gate, the chosen three-qubit gate does not permit a reduced process matrix which can be efficiently evaluated. A more general strategy for obtaining estimates of the fidelity of a quantum gate is to use fidelity estimation through importance sampling [55]. In this work, zero-fidelity estimation [18] is used as a faithful approximation to the full process fidelity as it is well suited to implementations on NISQ hardware.

The zero-fidelity between a unitary target gate $U$ and a noisy experimental gate $\Gamma$ may be written

$$F_0(U,\Gamma) = \frac{1}{d^2} \sum_{i,j=1}^{d^2} \text{Tr}[U\rho_i U^\dagger W_j]\,\text{Tr}[\Gamma(\rho_i)W_j],\tag{16}$$

where the input states $\{\rho_i\}$ are formed as the tensor product of single qubit symmetric informationally complete (SIC) states [56] and the operators $\{W_j\}$ form an orthonormal basis.

The zero-fidelity can be efficiently approximated by sampling $l$ input state/measurement basis pairs from the joint probability distribution

$$\text{Pr}(i,j) = \frac{1}{d}\text{Tr}[U\rho_i U^\dagger W_j],\tag{17}$$

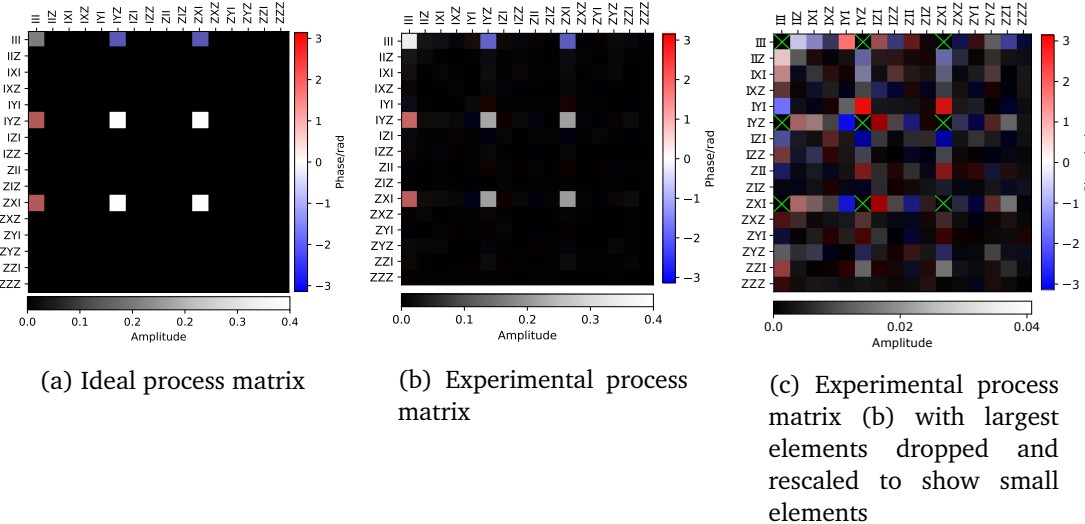

(a) Ideal process matrix

(b) Experimental process matrix

(c) Experimental process matrix (b) with largest elements dropped and rescaled to show small elements

Figure 6: Experimental results for the optimization of an $\exp(i\pi/4(ZX\mathbb{1} + \mathbb{1}YZ))$ gate: (a) shows the ideal process matrix for the gate (showing only elements that can be generated by Eq. (5)) and (b) shows the experimental process matrix for the optimal drive parameters obtained through BO, implemented on the `ibm_oslo` quantum device. (c) shows the same experimental results as (b) but with the 9 largest terms (corresponding to $\chi$ elements indexed by products of $\mathbb{1}\mathbb{1}\mathbb{1}, \mathbb{1}YZ$ and $ZX\mathbb{1}$) set to 0 (indicated by green crosses) and the color bar rescaled to show the magnitude of the less significant terms. Only terms that can be generated by the noisy cross-resonance and single qubit drive pulses are shown for clarity; the full process matrices including the dropped terms are shown in Appendix B. The qualitative features of the process matrix are accurately obtained, with the amplitude of the observed terms slightly reduced from the ideal matrix. Additionally, the Rabi angle is slightly misaligned, leading to a final process fidelity of 82%.

and for each experimental setting evaluating the estimator

$$X(i,j) = \frac{\mathrm{Tr}[\Gamma(\rho_i)W_j]}{\mathrm{Tr}[U\rho_i U^\dagger W_j]}, \tag{18}$$

for which the expected value is the zero-fidelity. The variance of this estimator is independent of the system size (although the individual expectation values still need an exponentially increasing number of projective measurements to be resolved) and converges to 0 as the zero-fidelity approaches unity; additionally, as the zero-fidelity increases, the difference between it and the process fidelity decreases. This makes the zero-fidelity well suited to optimization problems.

## 4.2 Optimization results

The final results for the optimization of the $\exp(i\pi/4(ZX\mathbb{1} + \mathbb{1}YZ))$ gate are shown in Fig. 6, with 6(a) showing the ideal target process matrix and 6(b) the experimental gate following the two part optimization protocol described in the previous section. While the optimization protocol was performed using zero-fidelity optimization with 200 expectation value measurements per estimation, the final result shown was obtained through full process tomography implemented on the `ibm_oslo` quantum device. Fig. 6(c) shows the same data with the 9 largest process matrix terms (corresponding to $\chi$ elements indexed by products of $\mathbb{1}\mathbb{1}\mathbb{1}, \mathbb{1}YZ$

and $ZX\mathbb{1}$) set to 0 (indicated by green crosses), with the rest of the matrix elements rescaled to allow the magnitude of the other process matrix terms to be seen. These elements have magnitudes of at most 0.04, showing that the dynamics are dominated by the 9 terms seen clearly in Fig. 6(b). Moreover, the magnitude of these terms is consistent with the level of measurement error in the device.

Only terms which are able to be generated from the application of the two unoptimized cross-resonance Hamiltonians Eq.(5) are shown. Appendix B shows the full process matrices for this experiment, in full form and with the 9 largest elements dropped. The full process matrix shows that all process matrix elements which are dropped from Fig. 6 have magnitudes smaller than the elements shown.

The final achieved process fidelity for the process matrix in Fig. 6 was 0.82, which is consistent with the achieved fidelities of the constituent $ZX\mathbb{1}$ and $\mathbb{1}YZ$ gates, both of which were approximately 90%. As above, these process fidelity values were obtained without state preparation and measurement error mitigation, and thus are underestimates of the quality of a gate as used in a quantum algorithm. With this in mind, the obtained fidelity is close to the optimal fidelity that can be achieved in the presence of measurement error – the fidelity of the three-qubit identity gate obtained using the same experimental protocol is 88%.

The obtained fidelity of 0.82% for this gate could represent a significant improvement in utility for NISQ devices, since the gate-based implementation requires substantially more pulses per two-body interaction and since the overall three qubit gate must be composed from the two-body interactions through Trotterization [57], further increasing the gate overhead.

## 5 Towards an engineered three-body gate

The previous sections demonstrate that VQGO can be used to obtain high fidelity two and three qubit gates. This could allow for the realization of more efficient hybrid quantum computations implemented on FF transmon devices, with computations composed into a wider set of basis gates, each of which may be obtained through VQGO. A natural question arises as to how far the protocol can be pushed: can the optimization be used to obtain a high-fidelity Floquet-engineered interaction for example [58, 59]?

Floquet engineering uses periodic driving fields to realize a time-dependent, periodic Hamiltonian $H(t + T) = H(t)$. Using Floquet theory [27, 60, 61], the propagator for this system can be expressed as $U(t) = U_F(t)\exp(-iH_F t)$ in terms of a *time-independent effective Hamiltonian $H_F$* and a periodic micromotion operator $U_F(t + T) = U_F(t)$. This is achieved by expressing $H(t)$ in the rotating frame defined by the micromotion operator,

$$H_F = U_F^\dagger(t)H(t)U_F(t) - i\dot{U}_F^\dagger(t)U_F(t). \tag{19}$$

At integer multiples of the driving period $t = nT$, the micromotion operator reduces to the identity and the dynamics of the system are entirely captured by the time-independent effective Hamiltonian as $U(nT) = \exp(-iH_F nT)$. By making use of Floquet engineering and using $U(nT)$ as a target gate, the range of gates which may be implemented on a given device can be expanded.

In this section, the results of the implementation of a Floquet-engineered three-body $\exp(i\theta ZYZ)$ gate based on an existing theoretical drive scheme [28–31] are presented. The extension to full time-dependent quantum control implies a significant increase in difficulty due to the increase in parameters and due to the precision in control parameters required to realize Floquet-engineered dynamics. As a result, the experimental implementation of this protocol represents an evaluation of the limitations of the VQGO routine as implemented on the IBM Quantum devices.

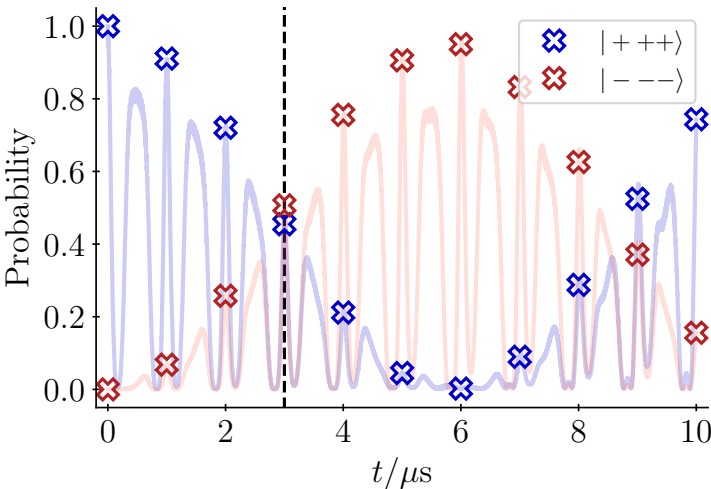

Figure 7: Evolution of the population of states $|+++\rangle$ (blue) and $|---\rangle$ (red) evaluated numerically using a full transmon model, demonstrating the Floquet-engineered $ZYZ$ interaction. Symbols (empty crosses) indicate the stroboscopic evolution in steps of $T = 1\mu s$, while solid shaded lines show the micromotion. At the third Floquet period ($t = 3T = 3\mu s$, indicated by the dashed black line), the three-qubit interaction realizes a beamsplitter operation between the two states, producing a three-qubit entangled state.

## 5.1 Theoretical protocol

In the theoretical driving protocol, a three-body interaction is predicted to appear in the presence of stable pairwise interactions as a second-order process arising from driving the central qubit in a chain of three coupled qubits. Concretely, the protocol assumes a static drift Hamiltonian

$$H_0 = J_{ZX}\left(ZX\mathbb{1} + \mathbb{1}XZ\right),\tag{20}$$

which may be modulated by a single qubit drive pulse of the form $\frac{\Omega(t)}{2}\mathbb{1}Y\mathbb{1}$. Optimal parameters $\Omega_k$ can be numerically found such that a modulation

$$\Omega(t) = \sum_{k=0}^{2}\Omega_k\cos(k\omega t),\tag{21}$$

produces the desired interaction at multiples of the fundamental Floquet driving period $T = 2\pi/\omega$.

Optimal parameters for this scheme were obtained in Ref. [31] based on an idealized Hamiltonian and are presented in Table 1 in Appendix A. To verify that these parameters remain optimal when moving from always-on, static interactions to a transmon system in which the interactions are dynamically switched on, numerical simulation of the protocol is performed. For most of the experimental parameters, realistic values based on experimental devices are chosen as outlined in Appendix A. For the two-body coupling term $J_{ZX}$, a compromise had to be made between obtaining a gate as quickly as possible and avoiding transitions to unwanted energy levels. The choice of $J_{ZX}/2\pi = 0.2$ MHz was found to be the optimal choice, which in turn fixes $\omega/2\pi = 1$ MHz, taking the $\omega = 5J_{ZX}$ case in Ref. [31], and $T = 1\,\mu s$. This results in a three-body strength of $J_{ZYZ}/2\pi = 0.04$ MHz, which yields an almost maximally entangling $ZYZ$ gate after three Floquet periods ($6\pi/25 \approx \pi/4$).

The resulting characteristic dynamics for an initial state $|+++\rangle$ is shown, as an example, in Fig. 7. The three-qubit interaction $ZYZ$ successfully induces Rabi oscillations between

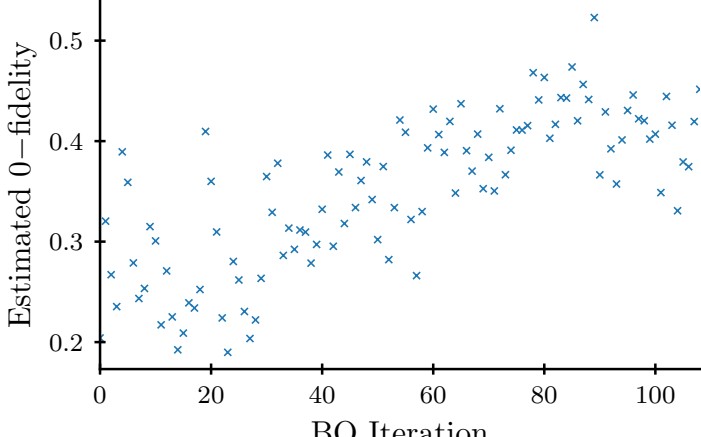

Figure 8: Convergence plot for the Bayesian optimization of a three-body $\exp(-6\pi/25\,ZYZ)$ interaction showing the estimated zero-fidelity as a function of Bayesian optimization iteration. Although the optimization is able to make modest improvements to the zero-fidelity, converging to an estimated zero-fidelity of approximately 0.4 (here the fluctuations are largely due to the non-zero variance of the zero-fidelity estimation), the achieved fidelities are significantly lower than can be achieved for the gates based on static interactions.

states $|+++\rangle$ and $|---\rangle$ at stroboscopic times, realizing to a good approximation a maximally entangling gate at time $t = 3T$, indicated by the black dashed line in Fig. 7. These simulations provide strong evidence that the three-qubit interaction should be observable in the experiment.

## 5.2 Optimization results

In order to minimize the overhead of the optimization protocol, it is useful once again to preoptimize the individual two-body $ZX\mathbb{1}$ and $\mathbb{1}XZ$ interactions such that they are high-fidelity and have interaction strengths as close to one another as possible. Once these terms are optimized, the weights of the single qubit driving parameters $\{\Omega_k\}$ and the amplitude of the compensating $\mathbb{1}X\mathbb{1}$ pulse may then be used as the optimization parameters. As with the previous two qubit gate, no convenient reduced process matrix can be generated from the drive protocol, and so zero-fidelity estimation was used as the figure of merit. As motivated in the previous section, the chosen Rabi angle for this interaction was $6\pi/25$, which is close to a maximally entangling gate whilst conforming to the requirement that the simulation time be an integer multiple of the Floquet period.

As shown in Fig. 8, as the optimization progresses the observed estimated zero-fidelities increase and fewer low fidelity results are observed, indicating that the optimizer is adapting to the parameter landscape. The spread of the data points even after approximately 80 iterations can largely be ascribed to the non-zero variance of the zero-fidelity estimation. Despite these modest improvements, the achieved fidelity is significantly lower than is observed for the previous gates, with the optimizer converging to an estimated zero-fidelity of approximately 0.4. In order to perform the optimizations, experimental jobs must be submitted to a queueing system to be implemented on the physical hardware. This can increase the necessary time for an optimization significantly. For the three-body gate, the optimization time was approximately 12 hours. Over this duration, the parameters characterizing the device can drift. For the static gates optimized in the previous section, this is unproblematic, since small drifts induce only

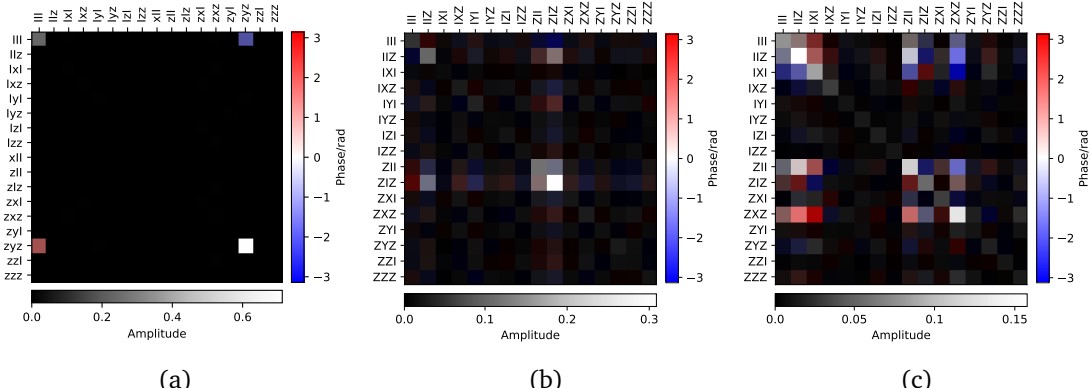

Figure 9: (a) ideal process matrix for the $\exp(-i\theta ZYZ)$ target unitary gate. (b) process matrix associated with the pulse parameters that yielded the highest estimated zero-fidelity, evaluated using full process tomography implemented on the `ibmq_jakarta` quantum device. (c) process matrix submitted to the experimental queue immediately following (b), finishing after approximately 8 hours, with the same pulse parameters. Only process matrix elements generated by (5) are shown, with the full process matrices given in Appendix B. Not only are both optimized process matrices extremely different from the ideal case, the last two are significantly different from each other, showing that drift in the machine is a substantial problem. It is likely that this is the reason for the VQGO is not able to reach similarly high fidelities to the previous gates.

small changes to the effective Hamiltonian, resulting in, for example, an over-rotation error. These errors can be effectively handled by the optimizer and so high fidelity results are still achievable. However, for the Floquet-engineered system a modest drift can induce significant changes in the effective Hamiltonian since the scheme requires precise cancellation of the nested commutators of the drive Hamiltonian terms. This can be intuitively observed in Fig. 7: very small shifts away from the stroboscopic drive time induce large deviations from the desired $ZYZ$ dynamics.

To investigate whether parameter drift is consistent with the experiment as an explanation for the difference between the static and Floquet VQGO schemes, quantum process tomography can be used. By repeating the process tomography twice with the exact same pulse setup, the effects of parameter drift can be observed. Fig. 9 shows the results of such a pair of experiments, with Fig. 9a showing the ideal process matrix and where the experiments that generated Fig. 9c were completed approximately 8 hours following the completion of the experiment presented in Fig. 9b. The drive parameters that were chosen correspond to the optimal parameters obtained from the VQGO routine. In all cases, only matrix elements which are generated by the Hamiltonian Eq.(5) are shown, with the other terms being significantly lower in magnitude. The full experimental process matrices are given in Appendix B.

Neither experimental process matrix reproduces the desired dynamics, even qualitatively, which is to be expected from the low achieved fidelity during the optimization. The deviation from Fig. 9b to Fig. 9c is much more significant: The quantum gates generated by the two pulse schemes are completely different, despite the fact that they were implemented with identical pulse parameters with less than a day between the experiments. This indicates that parameter drift is indeed a significant problem for obtaining high fidelity Floquet-engineered gates using VQGO.

Even with the limitations imposed by the noise levels of current NISQ devices, VQGO works well for optimizing gates based on static Hamiltonians. However, for time dependent

Hamiltonians the requirements for obtaining high fidelity gates are much higher, being beyond the reach of current devices. It may be possible to design control routines that are more robust to parameter drift such that these requirements are relaxed enough that VQGO can effectively realize high fidelity gates.

# 6 Conclusions

Given the high noise rates characteristic of NISQ devices, finding more efficient methods of implementing quantum algorithms could be a potentially valuable route towards obtaining useful experimental results in the near term. Variational quantum gate optimization (VQGO) is a protocol that can be used to obtain high fidelity gates in the presence of experimental noise and could therefore be used to expand the utility of NISQ devices.

In this work, we propose a VQGO protocol which uses the native operations of a given quantum device to obtain high fidelity gates efficiently. We experimentally evaluate the protocol through an implementation on fixed-frequency, fixed-interaction transmon qubits.

VQGO is shown to be highly effective at obtaining high-fidelity quantum gates based on static effective Hamiltonians. For a two-qubit maximally entangling $ZX$ target gate, VQGO is able to obtain pulse parameters which yield a process fidelity of 93% and for a three qubit gate a fidelity of 83% is achieved. These are very promising results, since the two qubit gate fidelity matches the IBM-optimized gates used for implementing CNOT gates, using fewer pulses and shorter total pulsing time, and since both sets of fidelities are very close to that of the identity gate (95% and 88% for the two and three qubit experimental identity gates respectively), which indicates the upper bound on achievable fidelity without measurement error mitigation. As part of the optimization protocol, we derive a reduced process matrix for the two-qubit gate which may be experimentally evaluated using only 12 measurements and apply zero-fidelity estimation as the figure of merit for the three qubit gate.

We assess the limitations of the scheme through an extension to the optimization of a Floquet-engineered, time-dependent gate. While the VQGO protocol is able to increase the fidelity of the implemented gate, the increased requirements of the time-dependent scheme combined with significant parameter drift over the duration of the experiment prevent the protocol from reaching similarly high fidelities to the gates based on static Hamiltonians. It is possible that driving schemes which are robust to this drift could be engineered. However, currently VQGO on FF tranmon qubits is only effective for target gates based on time-independent Hamiltonians.

# 7 Outlook

In this work, VQGO is shown to be capable of obtaining high fidelity gates based on static effective Hamiltonians. A direct application of VQGO is in quantum simulation. For many systems of interest, it is possible to obtain mappings to the native operations of a given device that are more efficient in terms of hardware resources than a decomposition into CNOT and single qubit rotation gates. An example of this is the transverse field Ising model, which can be mapped exactly to the native operations of FF transmon devices. By using VQGO to optimize blocks of Ising-like gates, the number of Trotter steps required to reach a given evolution time could be considerably reduced, expanding the reach of current devices for quantum simulation. Optimal decompositions into optimizable gates for a given system is therefore a valuable route for future work.

The target gates and figures of merit investigated in this work are specific to FF transmon qubits, but the general framework of VQGO is applicable to any system. It would thus be instructive to investigate the viability of VQGO on other NISQ systems. Zero-fidelity estimation can be applied to any quantum platform (and may be more efficient for certain platforms such as NMR quantum computers [18]) but the existence of reduced process matrices for systems other than FF transmon qubits warrants further investigation.

One of the advantages of using black-box optimization protocols to obtain optimal experimental parameters is that unknown errors in the device can be accounted for without a rigorous characterization of the physical device. Nevertheless, it could be valuable to complement the techniques outlined in this work with numerical and experimental characterization techniques in order to investigate the robustness of different pulse schemes. This could be used to inform which classes of parametrized pulses have the most potential for use in a VQGO scheme, expanding the utility of the protocol. The VQGO procedure proposed here may then also be adapted to optimize for the protocol robustness in response to variations of the optimal pulse parameters, besides the gate fidelity, by minimizing a suitably modified cost function.

An additional route for future work lies in addressing the difficulties associated with applying VQGO to the optimization of gates based on time-dependent Hamiltonians on FF transmon devices. It would be interesting to investigate whether more robust driving schemes that are stable with respect to moderate drifts in control parameters may be derived. It is possible that with an appropriate choice of driving routine, the utility of VQGO could be expanded to this regime. Having this limitation in mind when designing control schemes could lead to creative solutions which have not yet been considered – for instance, a control scheme could be developed that approximately realizes a given target gate over a range of parameters, as opposed to schemes which exactly realize a target gate but only for a precise configuration of control parameters.

## Acknowledgments

We are grateful to Adam Smith and Daniel Malz for providing stimulating discussions and to Marin Bukov for helpful comments on the manuscript.

**Funding information** This work is supported by Samsung GRP grant, the UK Hub in Quantum Computing and Simulation, part of the UK National Quantum Technologies Programme with funding from UKRI EPSRC grant EP/T001062/1 and the QuantERA ERA-NET Co-fund in Quantum Technologies implemented within the European Union's Horizon 2020 Programme. S.G. is supported by a studentship in the Quantum Systems Engineering Skills and Training Hub at Imperial College London funded by EPSRC (EP/P510257/1). F.P. acknowledges support from the Deutsche Forschungsgemeinschaft (DFG) via the Research Unit FOR 2414 under Project No. 277974659. We acknowledge the use of IBM Quantum services for this work. The views expressed are those of the authors, and do not reflect the official policy or position of IBM or the IBM Quantum team.

## A   Simulations of the device dynamics

In this appendix, we discuss how numerical simulations of the device have been performed, and the related parameters used.

The three qubits are described by the Hamiltonian

$$H(t) = \sum_{j=1}^{3} H_{Q_j} + H_{\text{int}} + \sum_{j=1}^{3} H_{\text{drive},j}(t). \tag{A.1}$$

The transmon Hamiltonians $H_{Q_j}$ read [25,62]

$$H_{Q_j} = \frac{\omega_{h,j}}{4}\left[\hat{y}_j^2 - \frac{2}{\epsilon_j}\cos(\sqrt{\epsilon_j}\hat{x}_j)\right], \tag{A.2}$$

where $\hat{y}_j = -i(\hat{b}_j - \hat{b}_j^\dagger)$ and $\hat{x}_j = \hat{b}_j + \hat{b}_j^\dagger$. The bosonic operators $\hat{b}_j$ describe (in unitless form) zero-point-fluctuations of transmon flux and charge. The harmonic frequencies $\omega_{h,j}$ and the anharmonicities parameters $\epsilon_j$ used are chosen to give transmon level splittings in the range of typical IBM qubits. In particular, they are fine tuned to give first-excitation frequencies 5.236 GHz, 5.014 GHz and 5.178 GHz and anharmonicities −0.340 GHz, −0.342 GHz and −0.341 GHz for qubits $Q_1$, $Q_2$, and $Q_3$, respectively. The transmon-transmon interaction Hamiltonian and the drive Hamiltonians read

$$H_{\text{int}} = \sum_{j=1,3} J_j \hat{y}_j \hat{y}_2, \tag{A.3}$$

$$H_{\text{drive},j}(t) = \Omega_j(t)\sin(\omega_j t - \phi_j)\hat{y}_j. \tag{A.4}$$

The couplings $J_j$ are chosen as $J_1/2\pi = 1.955$ MHz and $J_3/2\pi = 2.052$ MHz. State propagation in simulations is done by including 64 lowest-energy states in the composite Hilbert space, which gives converged results for the eight-dimensional three-qubit subspace. Choosing phases $\phi_j = (\pi, \pi/2, \pi)$, the Hamiltonian (A.1) is predicted to yield the following effective Hamiltonian in the qubit subspace, in a frame rotating at the qubit frequencies,

$$\begin{aligned}H_{\text{qub}} = {}& c_{Z\mathbb{1}\mathbb{1}}Z\mathbb{1}\mathbb{1} + c_{\mathbb{1}\mathbb{1}Z}\mathbb{1}\mathbb{1}Z + c_{ZZ\mathbb{1}}ZZ\mathbb{1} + c_{\mathbb{1}ZZ}\mathbb{1}ZZ \\ & + c_{\mathbb{1}X\mathbb{1}}\mathbb{1}X\mathbb{1} + c_{ZX\mathbb{1}}ZX\mathbb{1} + c_{\mathbb{1}XZ}\mathbb{1}XZ + \Omega_2(t)\mathbb{1}Y\mathbb{1}/2.\end{aligned} \tag{A.5}$$

The drive $\Omega_2(t)$ is chosen of the form (21) and is used to produce the target three-body interaction, as discussed in Section 5, in such a rotating frame. The Hamiltonian (A.5) contains additional terms as compared to the ideal two-body Hamiltonian $H_0$ (Eq. (20) of Sec. 5), which could potentially hinder the desired three-qubit effect. The terms $ZZ\mathbb{1}$ and $\mathbb{1}ZZ$ are weaker than other terms by an order of magnitude. They thus contribute little to the dynamics and can be neglected. The term $c_{\mathbb{1}X\mathbb{1}}\mathbb{1}X\mathbb{1}$ needs active compensation instead, while aiming at attaining the same magnitude of $ZX\mathbb{1}$ and $\mathbb{1}XZ$. This is done by introducing an additional drive $-\Omega_c\sin(\omega_2 t - \pi)\hat{y}_2$. The compensating amplitude $\Omega_c$ is calibrated empirically by inspecting simulated Rabi oscillations between states $|0\rangle$ and $|1\rangle$ of the central qubit, when driving either only $Q_1$ or $Q_3$. Indeed, when driving $Q_1$, these oscillations should occur at rate $r_0 = c_{\mathbb{1}X\mathbb{1}} + c_{ZX\mathbb{1}} - \Omega_c/2$ if $Q_1$ is in state $|0\rangle$, and at rate $r_1 = c_{\mathbb{1}X\mathbb{1}} - c_{ZX\mathbb{1}} - \Omega_c/2$ if $Q_1$ is in state $|1\rangle$. We iteratively search for a value of $\Omega_c$ yielding $|r_0| = |r_1|$ for both $ZX\mathbb{1}$ and $\mathbb{1}XZ$, and then adapt $\Omega_1$ and $\Omega_3$ until $c_{ZX\mathbb{1}} = c_{\mathbb{1}XZ}$ is also obtained. The terms $Z\mathbb{1}\mathbb{1}$ and $\mathbb{1}\mathbb{1}Z$ commute with the drive on the central qubit and the target $ZYZ$ interaction, and thus they do not interfere with the Floquet engineering scheme. Once their magnitude is determined, they can be included in the rotating frame or corrected with an initial pre-rotation of the qubits. To determine their magnitude, we proceed similarly to the case of $\mathbb{1}X\mathbb{1}$, namely we study the imbalance in the Rabi oscillations of $|++0\rangle$ and $|+-0\rangle$ for $Z\mathbb{1}\mathbb{1}$, and $|0++\rangle$ and $|0+-\rangle$ for $\mathbb{1}\mathbb{1}Z$. All (rounded) parameter values used as an example for the transmon Hamiltonian are summarized in Table 1.

Table 1: Simulation parameters for Fig. 7. All dimensionfull quantities are expressed in MHz.

| $\omega_{h,j}/2\pi$ | $\epsilon_j$ | $J_j/2\pi$ |
|---|---|---|
| $[5544, 5323, 5486]$ | $[209, 218, 212]$ | $[1955, 2052]$ |
| $\Omega_1/2\pi$ | $\Omega_3/2\pi$ | $\Omega_c/2\pi$ |
| 18.24 | 19.76 | 0.466 |
| $\Omega_{2k}/2\pi$ | $\omega/2\pi$ | |
| $[0.080, 2.170, 2.491]$ | 1.000 | |

While the parameter search for the compensating pulses is already successfully attained 'by hand' in simulations, in the experiment it is done via Bayesian optimization based on the noisy experimental data, as discussed in the main text, according to the VQGO algorithm proposed in this work.

# B  Full three qubit process matrices

For the results presented in Secs. 4 and 5, the process matrices are more conveniently expressed in a reduced form in which elements that cannot be generated from error terms in the cross-resonance and resonant drives are dropped. As evidence that this is indeed a good approximation, in this appendix the full three qubit process matrices for these results are given. Only the experimental gates are shown here: The ideal gates are numerically generated and so dropped elements are 0 up to floating point error.

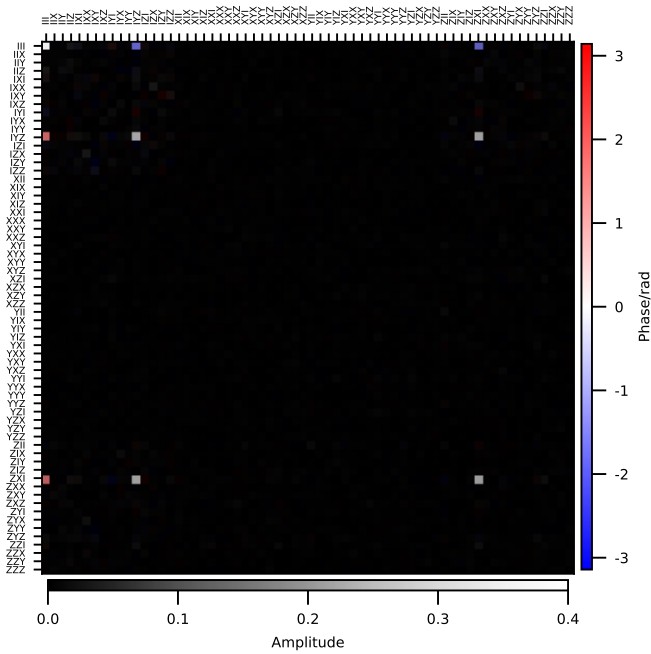

Figure 10: Full three qubit process matrix for the experimental data presented in Fig. 6. On this scale, no significant matrix elements other than those presented in Fig. 6(b) are observed.

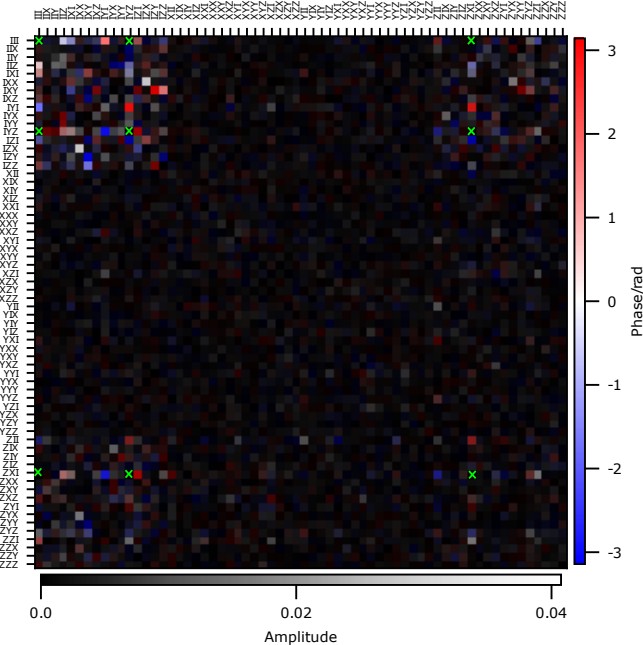

Figure 11: Full three qubit process matrix for the experimental data presented in Fig. 6 with the 9 largest elements set to 0 so that the magnitude of the smaller elements can be observed. Similarly to Fig. 6(c), the magnitude of the other elements is $\lesssim 0.04$, significantly lower than the principle terms in the full process matrix. Additionally, the terms included in Fig. 6(c) are much larger than the dropped terms.

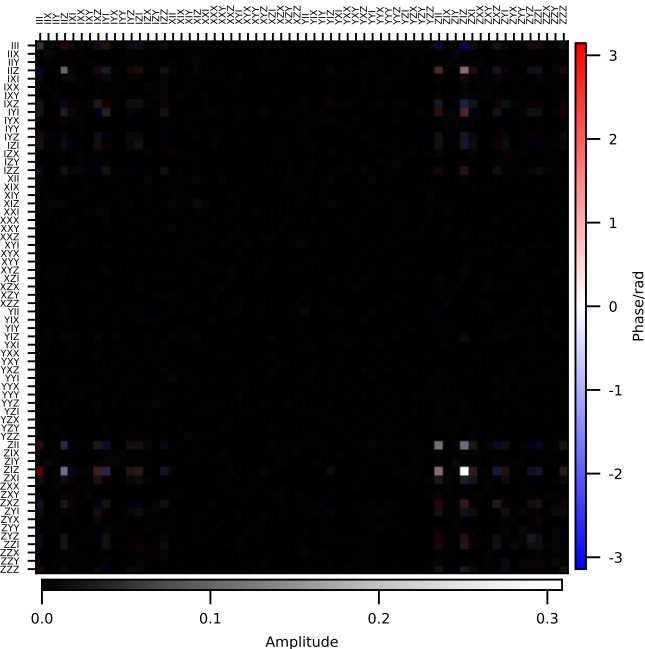

Figure 12: Full three qubit process matrix for the experimental data presented in Fig. 9a(b). Terms included in Fig. 9a(b) are much larger than the dropped terms.



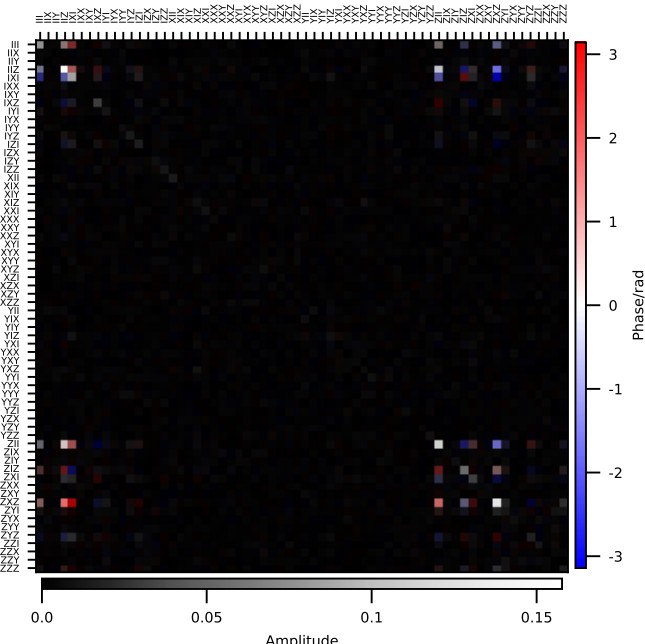

Figure 13: Full three qubit process matrix for the experimental data presented in Fig. 9a(c). Terms included in Fig. 9a(c) are much larger than the dropped terms.

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
