# Peer review of "Variational Quantum Gate Optimization at the Pulse Level"

_SciPost Physics, doi:SciPost Phys. 16, 082 (2024)_

## Round 2 · Referee Report · Anonymous (Referee 1) · 2023-5-2

Strengths

1) Straightforward application of pulse optimization on a quantum computer 2) Explore the possibility to create 2 and 3 qubit gates without complicated decomposition into small quantum gates. 3) Discuss the limit of the protocol 4) Use interesting figure of merit/cost functions that are adapted to quantum computers constraints.

Weaknesses

1) Poor comparison with the state-of-the-art quantum gates 2) The robustness of the pulses is not discussed

Report

The paper investigates the generation of quantum gates by optimizing the pulse shape directly on a quantum computer, which is ultimately an interesting thing, since we are not limited by the idealized model system, which can deviate more or less from the real experimental setup.

The paper focuses on quite complicated quantum gates, 2-qubit and 3-qubit gates, which are obviously typical systems where the method can reveal its full potential.

A critical point of view is applied, and an example where the optimization procedure fails is detailed, showing that the optimization problem is not trivial, and further work seems necessary to arrive at a general optimization scheme.

The paper globally well written and with a good reading flow. The first part is pedagogical, and the subtleties concerning the choice of cost function are well presented.

However, I think that the analysis of the optimized pulses is quite limited, in the sense that they are only partially compared to the existing pulse sequence. This is particularly true at the end of Sec. IV where the performance gain of performance remains only a hypothesis. Would it be possible to provide a quantitative comparison in this case? Moreover, at the end Sec. III, the direct comparison with a CNOT gate is not very fair since $U_{ZX}(\pi/4)$ is not exactly a CNOT gate.

In addition, the comparison is limited, in all cases, to a study of the fidelity, but the duration of the control fields, and their robustness are also important data. Obviously, the robustness of the optimized pulses are not easily determined, but it may be possible to estimate by simulating the system (like the simulation described in Appendix A) for a wide range of system parameters , and to compute the loss of fidelity with from the initial parameters. Maybe the optimized pulses are in average more robust than the state-of-the-art ones (for instance, the optimized CNOT gates may have a fidelity of >=93% on a larger area than the CNOT gate with a fidelity of 95%). Such a robustness analysis requires quite an important additional work, but I encourage the author to consider the inclusion of this kind of analysis in the paper (following the proposed idea or any other smarter comparison method).

I suspect that improving the elements of comparison with existing control fields would not be necessarily bad for the optimized pulses, in fact, the results may be slightly better than expected.

In addition to these remarks, I have a side question: How complicated are the optimized control field? Are they very different from the state-of-the-art ones? It could be nice to have a graph showing the difference between optimized and non-optimized control fields.

Concerning the validity of the work, I'm not able to reproduce the results in a reasonable amount of time, but the results look good. Sharing parts of the code used for the optimization (e.g., as a supplementary material) could be interesting for the community.

Requested changes

1) Provide a fair and quantitative comparison between state-of-the-art solution(s) for the optimized pulse discussed in Sec. III and IV. 2)Provide an additional element of comparisons for the pulses. The pulse duration seems an important and interesting quantity, and (optionally) the robustness of the pulses.

---

## Round 2 · Referee Report · Anonymous (Referee 2) · 2023-6-6

Report

Please find here enclosed my review of the manuscript entitled "Variational quantum gate optimization at the pulse level", submitted for publication in SciPost in Physics.
The authors investigate a variational quantum gate optimization protocol. Experimental optimization of two and three qubit gates is performed with the IBM quantum experience. The limits of the strategy are discussed and shown on a specific example.

The description and the understanding of fundamental limits in quantum control is a subject of fundamental interest and a basic prerequisite for applications in quantum computing and more generally in quantum technologies. In a closed-loop framework, the authors are able to optimize gate parameters that maximize the fidelity and are robust against noise effect, unwanted interactions or experimental imperfections. The proposed method and the results are interesting. The results of the paper seem sound and the paper is well-written. I support the publication of this paper. Optimization procedure is a very active area in quantum technologies with many different applications. In order to complete the bibliography of the paper, I suggest the authors to cite at least a recent review paper on the subject (1).

(1)- C. P. Koch et al., "Quantum optimal control in quantum technologies. Strategic report on current status, visions and goals for research in Europe", EPJ Quantum Technology volume 9, 19 (2022) [DOI: https://doi.org/10.1140/epjqt/s40507-022-00138-x]

---

## Round 3 · Author Response

Dear SciPost Editors,
Thank you very much for the invitation to respond to the reports of the two referees. We were happy to see both reviewers gave a highly positive assessment of our work, and hope our revised manuscript is now suitable for publication. It was particularly encouraging to read that our work is "of fundamental interest" (reviewer 1) and that they appreciated our "critical point of view" (reviewer 2). We thank the referees for their time in carefully reviewing and commenting on our work, and we believe thoroughly addressing their points has strengthened our manuscript.
In response to common comments from the referees, we have added an additional figure showing clearly the pulse parameters over which the optimizations were performed and we performed an additional experiment in order to more fairly compare our results against the current state-of-the-art techniques, as requested by reviewer 2. We have also expanded upon our explanations of our results, making explicit reference to other relevant parameters such as the pulse durations. It was also suggested (albeit optionally) that we could perform a more thorough analysis of the robustness of the pulses -- we argue that performing such an analysis properly is more appropriately left to a follow-up work, and so we have highlighted this in the Outlook section of the manuscript. A detailed breakdown of our responses to all the reviewer comments is attached.
We look forward to your response.
Yours sincerely,
Sean Greenaway, Francesco Petiziol, Hongzheng Zhao and Florian Mintert
Thank you very much for the invitation to respond to the reports of the two referees. We were happy to see both reviewers gave a highly positive assessment of our work, and hope our revised manuscript is now suitable for publication. It was particularly encouraging to read that our work is "of fundamental interest" (reviewer 1) and that they appreciated our "critical point of view" (reviewer 2). We thank the referees for their time in carefully reviewing and commenting on our work, and we believe thoroughly addressing their points has strengthened our manuscript.
In response to common comments from the referees, we have added an additional figure showing clearly the pulse parameters over which the optimizations were performed and we performed an additional experiment in order to more fairly compare our results against the current state-of-the-art techniques, as requested by reviewer 2. We have also expanded upon our explanations of our results, making explicit reference to other relevant parameters such as the pulse durations. It was also suggested (albeit optionally) that we could perform a more thorough analysis of the robustness of the pulses -- we argue that performing such an analysis properly is more appropriately left to a follow-up work, and so we have highlighted this in the Outlook section of the manuscript. A detailed breakdown of our responses to all the reviewer comments is attached.
We look forward to your response.
Yours sincerely,
Sean Greenaway, Francesco Petiziol, Hongzheng Zhao and Florian Mintert

---

## Round 3 · List of Changes

Referee 1:
We thank the referee for their time in reviewing our submission. We were pleased to read their positive
assessment of our work and were particularly pleased to read that “the results are interesting” and “the
paper is well-written”. We have added the suggested reference to the introduction in order to complete the
bibliography as suggested.
Referee 2:
We are grateful to the referee for their detailed review and we are pleased to read that the reviewer recognizes
the ”critical point of view” we are seeking to present and that “the paper is globally well written and with a
good reading flow”. We appreciate the suggestions for improving the paper and we believe that addressing
these comments has strengthened it significantly. In the sections below we address each of the comments
individually, highlighting changes to the manuscript where appropriate.
Referee comment:
“However, I think that the analysis of the optimized pulses is quite limited, in the sense that they are only
partially compared to the existing pulse sequence. This is particularly true at the end of Sec. IV where the
performance gain of performance remains only a hypothesis. Would it be possible to provide a quantitative
comparison in this case? Moreover, at the end Sec. III, the direct comparison with a CNOT gate is not very
fair since UZX (π/4) is not exactly a CNOT gate.”
Response:
We agree that such a comparison would strengthen the results presented in the paper and we
thank the referee for highlighting it. The quantitative comparison with the state-of-the-art pulse sequence
is possible, since we can pull the standard ZX echoed pulse sequence from the CNOT gate definition. We
performed exactly this experiment, obtaining a fidelity of approximately 93%, matching our results. This is
a favorable result, since our method has several advantages over the standard protocol:
• We use fewer pulses to achieve the same fidelity.
• We can simultaneously drive multiple interactions on different (but connected) qubits. This allows us
to implement the ZXI + IYZ gate in Sec. IV, which is not possible using the standard drive scheme.
• Our drive pulse achieves this fidelity with a shorter total pulse duration, meaning that one would be
less limited by dephasing and decoherence using our scheme than the standard method.
In addition to this, we also measured the process fidelity of the standard CNOT gate, which had a very
similar gate fidelity (again, approximately 93%). This suggests that we could use our scheme to generate
CNOT gates with shorter pulses, although we stress that our intended application is in the direct implemen-
tation of more complicated gates as explored in the paper.
We have added explanations of the above results to the new manuscript, please see the new paragraph
on page 7.
Referee comment:
“In addition, the comparison is limited, in all cases, to a study of the fidelity, but the duration of the
control fields, and their robustness are also important data. Obviously, the robustness of the optimized pulses
are not easily determined, but it may be possible to estimate by simulating the system (like the simulation
described in Appendix A) for a wide range of system parameters , and to compute the loss of fidelity with
from the initial parameters. Maybe the optimized pulses are in average more robust than the state-of-the-art
ones (for instance, the optimized CNOT gates may have a fidelity of ¿=93% on a larger area than the CNOT
gate with a fidelity of 95%). Such a robustness analysis requires quite an important additional work, but I
encourage the author to consider the inclusion of this kind of analysis in the paper (following the proposed
idea or any other smarter comparison method).”
Response:
We agree that fidelity alone does not capture the full picture of the results, so we have expanded our
discussion to capture other important features of the pulses. As mentioned in the previous section, we have
included a comparison with the standard method by which ZX interactions are induced in terms of both
fidelity and pulse duration. For the other gates, a direct apples-to-apples comparison is not available, since
those gates cannot be directly implemented using the standard gate set available on IBM devices (that is,
without using our method). In order to implement the three qubit gates, one would need to Trotterize the
target unitaries, adding an additional layer of error while expanding the number of gates required significantly.
We feel that this point could have been expressed more explicitly in the original manuscript, so we have
updated it for clarification.
For the robustness analysis, while we agree that such a discussion is important, we argue that a proper
analysis is better suited to a follow-up work. This is because a treatment of the topic would require a separate
extensive analysis, also in the case in which we would opt for performing only numerical simulations, without
experiments. Indeed, for meaningful simulations with the optimized pulses found in the experiment, one
would need first to formulate an error model reasonably reproducing the behaviour of the experimental
hardware, which is very challenging and bound to be approximate: this difficulty is the main motivation for
developing our proposed hardware-informed optimization scheme. An alternative could be to use a generic
error model and re-optimize the pulses based on multiple realizations of noisy simulated data. The resulting
pulses would however be of little use for the experiment, where errors will irremediably depart from such a
model. For instance, unknown distortions in the device would be difficult or impossible to account for, while
they are intrinsically dealt with in our VQGO algorithm. The above-mentioned analyses would then only
be preliminary steps, before proceeding with the actual robustness analysis, requiring extensive parameter
scans. We stress that a key feature of our work is that we essentially black-box these concerns and show that
one can obtain high fidelity gates in situ, without suffering for the above mentioned limitations.
We agree that our explanation of this in the original manuscript was lacking. In the updated manuscript,
we briefly discuss some aspects of robustness, such as the parameter regimes the optimization is expected to
be effective over, and we explicitly state that a rigorous robustness analysis would be a good path for future
work in the Outlook.
Referee comment:
“In addition to these remarks, I have a side question: How complicated are the optimized control field?
Are they very different from the state-of-the-art ones? It could be nice to have a graph showing the difference
between optimized and non-optimized control fields.”
Response:
We thank the referee for their question, as it highlights a positive feature of our work that was
not well presented in the original manuscript, namely the simplicity of the control fields used. As mentioned
previously, our pulse scheme uses fewer pulses than the standard IBM scheme to achieve a ZX gate. The
pulse shapes are also quite simple, being square pulses with Gaussian rise/falls at the beginning and end of
the pulses. Our optimization parameters are the amplitudes and phases of these pulses. This was not made
explicit enough in the original manuscript, and so we have added pulse diagrams and a further explanation
to the updated manuscript, please see the new Fig. 4 and the related discussion on page 5-6.
We thank the referee for their time in reviewing our submission. We were pleased to read their positive
assessment of our work and were particularly pleased to read that “the results are interesting” and “the
paper is well-written”. We have added the suggested reference to the introduction in order to complete the
bibliography as suggested.
Referee 2:
We are grateful to the referee for their detailed review and we are pleased to read that the reviewer recognizes
the ”critical point of view” we are seeking to present and that “the paper is globally well written and with a
good reading flow”. We appreciate the suggestions for improving the paper and we believe that addressing
these comments has strengthened it significantly. In the sections below we address each of the comments
individually, highlighting changes to the manuscript where appropriate.
Referee comment:
“However, I think that the analysis of the optimized pulses is quite limited, in the sense that they are only
partially compared to the existing pulse sequence. This is particularly true at the end of Sec. IV where the
performance gain of performance remains only a hypothesis. Would it be possible to provide a quantitative
comparison in this case? Moreover, at the end Sec. III, the direct comparison with a CNOT gate is not very
fair since UZX (π/4) is not exactly a CNOT gate.”
Response:
We agree that such a comparison would strengthen the results presented in the paper and we
thank the referee for highlighting it. The quantitative comparison with the state-of-the-art pulse sequence
is possible, since we can pull the standard ZX echoed pulse sequence from the CNOT gate definition. We
performed exactly this experiment, obtaining a fidelity of approximately 93%, matching our results. This is
a favorable result, since our method has several advantages over the standard protocol:
• We use fewer pulses to achieve the same fidelity.
• We can simultaneously drive multiple interactions on different (but connected) qubits. This allows us
to implement the ZXI + IYZ gate in Sec. IV, which is not possible using the standard drive scheme.
• Our drive pulse achieves this fidelity with a shorter total pulse duration, meaning that one would be
less limited by dephasing and decoherence using our scheme than the standard method.
In addition to this, we also measured the process fidelity of the standard CNOT gate, which had a very
similar gate fidelity (again, approximately 93%). This suggests that we could use our scheme to generate
CNOT gates with shorter pulses, although we stress that our intended application is in the direct implemen-
tation of more complicated gates as explored in the paper.
We have added explanations of the above results to the new manuscript, please see the new paragraph
on page 7.
Referee comment:
“In addition, the comparison is limited, in all cases, to a study of the fidelity, but the duration of the
control fields, and their robustness are also important data. Obviously, the robustness of the optimized pulses
are not easily determined, but it may be possible to estimate by simulating the system (like the simulation
described in Appendix A) for a wide range of system parameters , and to compute the loss of fidelity with
from the initial parameters. Maybe the optimized pulses are in average more robust than the state-of-the-art
ones (for instance, the optimized CNOT gates may have a fidelity of ¿=93% on a larger area than the CNOT
gate with a fidelity of 95%). Such a robustness analysis requires quite an important additional work, but I
encourage the author to consider the inclusion of this kind of analysis in the paper (following the proposed
idea or any other smarter comparison method).”
Response:
We agree that fidelity alone does not capture the full picture of the results, so we have expanded our
discussion to capture other important features of the pulses. As mentioned in the previous section, we have
included a comparison with the standard method by which ZX interactions are induced in terms of both
fidelity and pulse duration. For the other gates, a direct apples-to-apples comparison is not available, since
those gates cannot be directly implemented using the standard gate set available on IBM devices (that is,
without using our method). In order to implement the three qubit gates, one would need to Trotterize the
target unitaries, adding an additional layer of error while expanding the number of gates required significantly.
We feel that this point could have been expressed more explicitly in the original manuscript, so we have
updated it for clarification.
For the robustness analysis, while we agree that such a discussion is important, we argue that a proper
analysis is better suited to a follow-up work. This is because a treatment of the topic would require a separate
extensive analysis, also in the case in which we would opt for performing only numerical simulations, without
experiments. Indeed, for meaningful simulations with the optimized pulses found in the experiment, one
would need first to formulate an error model reasonably reproducing the behaviour of the experimental
hardware, which is very challenging and bound to be approximate: this difficulty is the main motivation for
developing our proposed hardware-informed optimization scheme. An alternative could be to use a generic
error model and re-optimize the pulses based on multiple realizations of noisy simulated data. The resulting
pulses would however be of little use for the experiment, where errors will irremediably depart from such a
model. For instance, unknown distortions in the device would be difficult or impossible to account for, while
they are intrinsically dealt with in our VQGO algorithm. The above-mentioned analyses would then only
be preliminary steps, before proceeding with the actual robustness analysis, requiring extensive parameter
scans. We stress that a key feature of our work is that we essentially black-box these concerns and show that
one can obtain high fidelity gates in situ, without suffering for the above mentioned limitations.
We agree that our explanation of this in the original manuscript was lacking. In the updated manuscript,
we briefly discuss some aspects of robustness, such as the parameter regimes the optimization is expected to
be effective over, and we explicitly state that a rigorous robustness analysis would be a good path for future
work in the Outlook.
Referee comment:
“In addition to these remarks, I have a side question: How complicated are the optimized control field?
Are they very different from the state-of-the-art ones? It could be nice to have a graph showing the difference
between optimized and non-optimized control fields.”
Response:
We thank the referee for their question, as it highlights a positive feature of our work that was
not well presented in the original manuscript, namely the simplicity of the control fields used. As mentioned
previously, our pulse scheme uses fewer pulses than the standard IBM scheme to achieve a ZX gate. The
pulse shapes are also quite simple, being square pulses with Gaussian rise/falls at the beginning and end of
the pulses. Our optimization parameters are the amplitudes and phases of these pulses. This was not made
explicit enough in the original manuscript, and so we have added pulse diagrams and a further explanation
to the updated manuscript, please see the new Fig. 4 and the related discussion on page 5-6.

---

## Editorial Decision

published